# Novel Characterization of Constipation Phenotypes in ICR Mice Orally Administrated with Polystyrene Microplastics

**DOI:** 10.3390/ijms22115845

**Published:** 2021-05-29

**Authors:** Yun Ju Choi, Jun Woo Park, Ji Eun Kim, Su Jin Lee, Jeong Eun Gong, Young-Suk Jung, Sungbaek Seo, Dae Youn Hwang

**Affiliations:** 1Department of Biomaterials Science (BK21 FOUR Program), College of Natural Resources & Life Science/Life and Industry Convergence Research Institute/Laboratory Animal Resources Center, Pusan National University, Miryang 50463, Korea; poiu335@naver.com (Y.J.C.); qcwalq@naver.com (J.W.P.); prettyjiunx@naver.com (J.E.K.); nuit4510@naver.com (S.J.L.); jegog@naver.com (J.E.G.); sbseo81@pusan.ac.kr (S.S.); 2College of Pharmacy, Pusan National University, Busan 46241, Korea; youngjung@pusan.ac.kr

**Keywords:** microplastics, polystyrene, constipation, stools, mucin, aquaporin

## Abstract

Indirect evidence has determined the possibility that microplastics (MP) induce constipation, although direct scientific proof for constipation induction in animals remains unclear. To investigate whether oral administration of polystyrene (PS)-MP causes constipation, an alteration in the constipation parameters and mechanisms was analyzed in ICR mice, treated with 0.5 μm PS-MP for 2 weeks. Significant alterations in water consumption, stool weight, stool water contents, and stool morphology were detected in MP treated ICR mice, as compared to Vehicle treated group. Also, the gastrointestinal (GI) motility and intestinal length were decreased, while the histopathological structure and cytological structure of the mid colon were remarkably altered in treated mice. Mice exposed to MP also showed a significant decrease in the GI hormone concentration, muscarinic acetylcholine receptors (mAChRs) expression, and their downstream signaling pathway. Subsequent to MP treatment, concentrations of chloride ion and expressions of its channel (CFTR and CIC-2) were decreased, whereas expressions of aquaporin (AQP)3 and 8 for water transportation were downregulated by activation of the mitogen-activated protein kinase (MAPK)/nuclear factor (NF)-κB signaling pathway. These results are the first to suggest that oral administration of PS-MP induces chronic constipation through the dysregulation of GI motility, mucin secretion, and chloride ion and water transportation in the mid colon.

## 1. Introduction

Due to increasing plastic wastes in oceans, MP have received great attention as pollutants of the marine environments [1]. Although MP are consumed by marine organisms, they progressively occur in nutrients from lower to higher organisms in the food chain, including mammals [2,3]. Recently, MP were evaluated in various cells and animals, and determined as one of the risk factors for human health. However, conflicting results were reported for the toxicity of MP against human cells [4]. Most previous studies showed that MP induce some degree of toxicity or pathological changes in human cells, whereas few studies suggest no significant cellular toxicity, except at high concentrations [5,6,7,8]. Alterations on several physiological responses, including oxidative stress, inflammatory cytokines secretion, cell cycle arrest, apoptosis, and histamine release, were detected in MP treated human cells [9,10,11]. Furthermore, the toxic and physiological effects of MP observed in in vitro experiments were similarly reflected in animal experiments. Most MP treatments induce various alterations in the toxicology and physiology of mice and rats, although changes were majorly accumulated in three major tissues, viz., liver, kidney, and gut [12,13]. Especially, numerous pathological changes in lipid metabolism, inflammation, lipid profile, and lipid accumulation, were observed in the liver tissue of MP treated animals [12,14,15]. Additionally, exposure to MP induces several immunological responses, such as secretion of interleukin (IL)-1α cytokine and T helper (Th) cells [16]. Conversely, no significant physiological responses, including tissue damage, inflammation, oxidative stress, and behavior, were induced by MP administration for 28 days in mice, or for 5 weeks in Wistar rats [17,18].

Meanwhile, a few strong findings have been presented to explain the correlation be-tween MP treatment and induction of constipation. The oral administration of PS-MP (0.5 and 50 μm size) for 5 weeks induces a significant modification of the gut microbiota composition, which is perceived as decreased relative abundances of *α-Proteobacteria* and *Firmicutes* in feces. A decrease of mucus secretion was also detected in the gut of these mice, regardless of MP size [19]. A similar alteration was observed in pregnant mice treated with PS-MP; 14 bacterial types were significantly altered at the genus level, while the mucus secretion and the transcription level of genes related to these bacteria were de-creased after exposure to PS-MP [20]. Furthermore, PS-MP treatment for 5 weeks resulted in enhanced number of gut microbial species, bacterial abundance, and flora diversity in the C57BL/6 mice model, where serum concentrations of inflammatory cytokines, including IL-1α, Il-6, IL-9, and RANTES, were also significantly increased [16]. However, no study has evaluated the oral administration of PS-MP and its effect on the incidence of constipation diseases.

The current study investigates the pathological symptoms and molecular mechanism of constipation in PS-MP treated ICR mice, through analysis of stool parameters, histopathology, GI transit, GI hormone secretion, mucin secretion, chloride ion regulation, and water channel expression. Results of this study indicate that MP treatment is probably a novel cause for constipation, accompanied by decreased GI motility, mucin secretion, and ion/water channel expression in ICR mice.

## 2. Results

### 2.1. Physicochemical Properties of MP

To analyze the physicochemical properties, we measured the morphological features and actual size of MP using SEM and size analyzer. MP exhibited a circular shape of regular size (Figure 1a). The number distribution size and zeta potential of MP were determined to be 593.83 ± 7.53 d.nm and 35.98 ± 0.26 mV, respectively (Figure 1b).

### 2.2. Effects of MP Administration on the Feeding Behavior and Stool Parameters

We first investigated whether MP administration affects the feeding behavior and excretion parameters of ICR mice. To achieve this, alterations in food intake, water consumption, urine volume, and stool parameters were measured in ICR mice after treatment with three different doses of MP. Compared to the Vehicle control, the MP treated group showed significant enhancement of water intake. However, no significant changes were observed for food intake and urine volume (Figure 2a). Of the three stool parameters evaluated, the weight and water content of stools were significantly decreased in the MP treated mice, as compared to the Vehicle mice, whereas stool number was maintained constant (Figure 2b). Especially, stool morphology was remarkably altered after MP administration. The production rate of abnormal shaped stools, including small, short, and irregular type, was 1.77–2.3 times greater in the MP treated groups than the Vehicle treated group (Figure 2c). Taken together, these results suggest that MP administration successfully induces the defecation delay by enhancing the water intake as well as production rate of abnormal shaped stools.

### 2.3. Effects of MP Administration on the GI Motility and Intestinal Length

To investigate whether the defecation delay in MP treated mice is accompanied by alterations in the GI motility and intestinal length, the charcoal meal transit test and intestine length analyses were performed in ICR mice treated with MP for 2 weeks. A dose-dependent decrease was observed in propulsion of the charcoal meal in the MP treated group, as compared to the Vehicle treated group. A similar pattern was observed for intestinal length, although it was not dose-dependent (Figure 3a,b). These results indicate that the MP-induced defecation delay is tightly associated with the dysregulation of GI motility and decrease in intestinal length.

### 2.4. Effects of MP Administration on the Histopathological and Cytological Structure in Mid Colon of ICR Mice

We investigated the associated changes in the histopathological and cytological structure of the mid colon, caused by the defecation delay in MP treated ICR mice. To achieve this, alterations in the hematoxylin and eosin (H&E)-stained histopathological structures and transmission electron microscopy (TEM) obtained ultrastructure were analyzed in the mid colon of subset groups. The thicknesses of mucosa, muscle, flat luminal surface, and crypt layer were significantly decreased in the MP treated group, as compared to the Vehicle group. Most of these decreases exhibited a dose-dependent pattern (Figure 4a). In addition, a similar pattern was detected in the number of goblet cells and crypts of Lieberkuhn. Subsequent to MP administration, these levels were lower than levels obtained in the Vehicle treated group, although a dose-dependent decrease was observed only in the number of goblet cells (Figure 4b). Moreover, the associated changes in the ultrastructure of crypts were further determined by TEM analysis. Significant alterations were observed on the crypts of Lieberkuhn of the mid colon. Goblet cells were inconsistent in shape and uneven in size after treatment with MP. Compared to the Vehicle group, the average number of mucus drops in each goblet cell was remarkably increased, and the number of dark vesicles was greater in Paneth cells of the MP treated group (Figure 5). These findings indicate that MP-induced defecation delay is associated with abnormalities in the histopathological and cytological structural of the mid colon.

### 2.5. Effects of MP Administration on the Concentration of GI Hormones in the Mid Colon

GI hormones play an important physiological role in regulating the smooth muscle contraction of the intestine [21]. To determine whether the MP-induced defecation delay is accompanied by alterations in the levels of GI hormones, the concentrations of cholecystokinin (CCK) and gastrin were measured in the mid colon of the Vehicle, LoMP, MiMP, and HiMP treated groups. Remarkable decreased was obtained in the concentrations of both CCK and gastrin in the mid colon of MP treated mice, as compared to the Vehicle treated mice. However, the gastrin concentration was maintained constant in the LoMP treated group (Figure 6a,b). These results indicate that MP-induced defecation delay is associated with the suppression of CCK and gastrin, which are involved in the regulation of intestinal muscle contraction.

### 2.6. Effects of MP Administration on the Downstream Signaling Pathway of mAChRs

Western blot analysis was performed to determine if the MP-induced defecation delay was accompanied by changes in the regulation of downstream signaling pathway of mAChRs. The expression levels of mAChR M2, mAChR M3, Gα, protein kinase C (PKC), p-PKC, phosphoinositide 3-kinases (PI3K), and p-PI3K protein were measured in the mid colons of all subset groups. The levels of mAChR M2 and mAChR M3 expression were dose-dependently and significantly decreased in the three MP treated groups, as compared to the Vehicle treated group (Figure 7a,b). However, their downstream signaling pathway exhibited a reverse pattern in the same groups. The levels of Gα expression, and PKC and PI3K phosphorylation were remarkably enhanced in the MP treated mice, except PKC phosphorylation in the LoMP treated group (Figure 7c,d). These results indicate that the MP-induced defecation delay is tightly associated with the dysregulation of mAChR expressions and their downstream signaling pathway in the mid colons of ICR mice.

### 2.7. Effects of MP Administrations on Mucin Secretion Ability in the Mid Colon

We next investigated whether the MP-induced defecation delays are accompanied by changes in the regulation of mucin secretion ability in the mid colon. To achieve this, the levels of mucin secretion and some related gene transcriptions were measured in the mid colon of the MP treated groups. In mid colons obtained from the Vehicle treated group, the goblet cells stained dark blue for mucin were constantly concentrated in the crypts of Lieberkühn. However, MP administration resulted in the rapid disruption and decreased intensity of these structures (Figure 8a). Moreover, these alterations detected in mucin staining analysis were completely reflected at the transcription level of three related genes. The transcription levels of the mucin 2 (MUC2), MUC1, and Kruppel-like factor 4 (Klf4) genes were lower in the MP treated groups than in the Vehicle treated group, although the decrease rates were widely varied (Figure 8b). Taken together, these results suggest that MP-induced defecation delay may be associated with the decreasing mucin secretion ability and transcription of mucin related genes in the mid colon.

### 2.8. Effects of MP Administration on the Regulation of Membrane Chloride ion Transport in the Mid Colon

To examine if MP-induced defecation delay is accompanied by changes in the regulation of chloride ion transport in the mid colon, the chloride ion concentration and its channel expressions were measured in the mid colon of the Vehicle, LoMP, MiMP, and HiMP treated groups. The concentration of chloride ion showed a remarkable dose-dependent decrease in the MP treated groups, as compared to the Vehicle treated group (Figure 9a). Also, the regulation pattern of chloride ion concentration was reflected at the transcription level of chloride channel genes. Expression levels of chloride channel 2 (CIC-2) and cystic fibrosis transmembrane conductance regulator (CFTR) mRNAs were significantly decreased in the LoMP, MiMP, and HiMP treated groups, as compared to the Vehicle treated group (Figure 9b). These results indicate that the defecation delay in MP treated groups is associated with dysregulation of the chloride ion transport in the mid colon.

### 2.9. Effects of MP Administration on the Regulation of Membrane Water Transport in the Mid Colon

Furthermore, we investigated whether the increase of water intake during MP-induced defecation delay is associated with the regulation of membrane water balance in the mid colon. Since AQP3 regulates the liquid water metabolic abnormalities and intestine permeability alteration via MAPK/NF-κB pathway, alterations in the AQP3 and AQP8 transcriptions and MAPK/NF-κB signaling pathway were examined in the mid colon of subset groups [22,23]. The mRNA levels of AQP3 and AQP8 were remarkably decreased in the LoMP, MiMP, and HiMP treated groups, as compared to the Vehicle treated group (Figure 10a). However, a reverse regulation pattern was observed in the MAPK/NF-κB signaling pathway that is involved in regulating the AQP transcription levels. Compared to the Vehicle group, phosphorylation levels of ERK, p38, NF-κB, and inhibitor of κB (IκB)-α proteins were dose-dependently and significantly increased in the MP treated groups (Figure 10b,c). These results indicate that decreasing transcription of membrane water channels via the activation of the MAPK/NF-κB signaling pathway probably contributes to the increase of water intake during MP-induced defecation delay.

### 2.10. Verification of MP Effects on the Regulation of Water and Chloride Transport in IEC18 Cells

Finally, we verified the effects of MP on the regulation of water and chloride transport in intestinal epithelial cells. To achieve this, the expression level of chloride channel and water transporter were measured in the IEC18 cells after MP treatment. Total cells in each group were maintained their morphology (Figure 11a). All analyzed factors including CIC-2, CFTR, AQP3 and AQP8 showed a similar alteration pattern that their transcription was lower in MP treated group than Vehicle treated group (Figure 11b,c). Also, alterations on the MAPK/NF-κB signaling pathway in MP treated IEC18 cells were compared with those of MP treated ICR mice. Activation of MAPK/NF-κB signaling pathway were commonly detected in IEC18 cells and transverse colon of ICR mice treated with MP (Figure 11d,e). Therefore, above results suggest that the effects of MP in mid colon of ICR mice are equally observed in epithelial cells.

## 3. Discussion

Blockages and nerve problems in the colon or rectum, as well as dysfunction of the smooth muscle and GI hormones, are some of the major causes attributing to chronic constipation in humans [24]. Similar phenotypes of this disease have been detected in animals after administration of various chemicals and drugs, including Lop [25], clonidine [26], morphine [27], opioid receptor antagonist [28], clozapine [29], and carbon [30,31]. However, no studies have investigated novel causes leading to chronic constipation, until now. The current study evaluates the possibility of PS-MP administration as a novel cause of chronic constipation. The results of this study provide scientific evidence that key phenotypes for chronic constipation are observed in ICR mice after oral administration of MP for 2 weeks. Our study further reveals that constipation detected in MP treated ICR mice is tightly linked to dysregulation of water consumption, stool morphology, GI motility, GI hormone concentrations, mAChR signaling pathway, and membrane transportation of ions and water. Especially, our study analyzed the mid colon of PS-MP treated ICR mice because it plays a key role in the fermentation of food matter, removal of water and nutrients, and formation of stools. Therefore, mid colon has been considered as important target for constipation studies in many previous studies [32,33,34].

MP treatment also induces some metabolic disorders in few specific organs, although they do not trigger serious chronic diseases. MP of varying sizes were the cause of the significant increase of oxidative stress in liver tissues [12,13,14]. Similar alterations were detected in acetylcholine esterase activity, lipid profile, energy metabolism, glycolipid metabolism, and lipid metabolism in the same tissue after MP treatment [12,13,19,35]. The induction of bile acid metabolic disorder and fatty acid metabolic disorder were observed in the gut of mice treated with PS-MP, and in the F1 offspring after exposure to the maternal PS-MP administered (0.5 and 5 μm) [15,35]. However, no study has evaluated the correlation between MP administration and other chronic diseases, including constipation, diabetes, and obesity. Therefore, the results of the present study demonstrate the first scientific evidence that MP administration is probably a major cause of chronic constipation, although further research is required to elucidate the molecular and cellular mechanisms of action.

In the current study, we examined alterations in the feeding behavior in ICR mice treated with three different doses of MP. Of the three feeding behavior parameters, only water consumption exhibits a dramatic change, although urine volume was altered but with no statistical significance. As shown in Figure 2a, water consumption was 2.3–3.5 times higher in the MP treated groups than the Vehicle treated group. These results are in partial agreement to previous studies, regardless of the inducing agents. In the activated carbon induced model, the amount of drinking water consumed remained constant during the early stages (from day 1 to day 7) and showed gradual decrease to 12% at a late stage of the experimental period (9 days) [31,36]. However, in the Lop induced constipation model, water consumption as well as food intake were maintained at a constant level in SD rats during the entire experimental duration [37,38,39]. We attribute these differences to the varied mechanistic actions of the treatment agents used in each study. Furthermore, our finding in water consumption provides some clues, that MP treatment is probably tightly linked to the dysregulation of water balance in the body.

The significant decrease in stool excretion is considered a key marker for constipation phenotypes in most studies, although the actual detection factors are varied in each study [25,37,40,41,42,43]. Previously, three stool-related factors, including stool number, weight, and water content, have been widely applied to evaluate laxative effects of therapeutic drugs in the constipation model [37,40]. The levels of these factors were remarkably decreased subsequently to treatment of Lop or carbon in the mice and rat model. None of these factors showed a differing pattern from the whole, and their levels in Lop or carbon treated animals were similarly maintained in most studies [25,31]. Experimental treatments with numerous therapeutic extracts revealed recovery of levels, although their recovery rates varied in each study [25,31]. However, in the current study, a different alteration pattern was observed in stool number after MP treatment, although the weight and water content of stools showed similar patterns as previous studies. The weight and water contents of stools were remarkably decreased after MP treatment, while the number of stools remained unaffected in the same group. However, significant changes were detected in the stool morphology. The number of abnormal shaped stools showed a 1.77–2.27 times increase in the MP treated groups, as compared to the Vehicle treated group (Figure 2c). Thus, the results of the present study provide the first evidence that MP-induced constipation is tightly correlated with morphological changes of stools, rather than the number of stools. These results provide an important clue for identifying the molecular mechanism involved in MP-induced constipation.

AQPs have received great attention as new therapeutic targets for treating constipation [44]. These proteins are small transmembrane proteins expressed in various cell types and play an important role in mediating the transmembrane water transport and regulation of GI fluid secretions [45,46]. AQPs are differentially distributed in the various cell types of the GI tract and are classified into two major groups: the “classical” water-permeable AQPs (including AQP1, 4 and 5), and water and glycerol-permeable AQPs (including AQP3 and 9) [47,48]. AQP3 and 4 are expressed in the basolateral membrane of epithelial cells of the GI tract, while AQP5 is distributed in the apical membrane. AQP7, 8, 10, and 11 are observed in the enterocytes of the intestine [49,50]. Among the several AQPs, AQP3 is strongly associated with constipation, although the expression is varied in different studies. Colons of the Lop treated model with slow transit constipation show increased levels of AQP3 expression [51], but a reverse pattern was detected in the constipation model. Moreover, significant decrease or down-regulation of AQP3 expression was observed in the colon of rat models with slow transit constipation [52,53]. Conversely, AQP3 expression levels were enhanced in the morphine-induced constipation model and morphine treated cancer patients with severe constipation [54,55]. The current study examined the expression level of AQP3 and downstream members of the NF-κB signaling pathway in mid colons of MP treated mice. Exposure to MP resulted in decreased transcription of AQP3 in the mid colon, via activation of the MAPK/NF-κB signaling pathway. The results of the present study in MP-induced constipation ICR mice showed partial agreement with previous results which reported that AQP3 expression was decreased or down-regulated in the colon of constipation rats. Furthermore, our results are the first to suggest a correlation between the alternative expression for the AQP3 and induction of constipation in the mid colon of MP treated ICR mice. However, further research is required to determine the molecular mechanism of action.

Meanwhile, only few studies were reported the role of MP on microbiota of mice. Treatment of polystyrene MP with 5 μm size for 6 weeks was induced by the change of the gut microbiota composition including 15 types of bacteria in ICR mice [15]. A similar response was observed on the gut microbiota including *Firmicutes* and *α-Proteobacteria* of ICR mice treated with two polystyrene MP for 5 weeks [19]. However, a significant increase on the number of gut microbial species, flora diversity, and bacterial abundance was detected in C57BL/6 mice model after treatment of polyethylene MP (10–150 μm) for 5 weeks [15]. Furthermore, the toxic effects of MP were investigated in various target organs including liver, kidney, testis, and ovary of mice and rat model. Among them, liver was analyzed as first target for toxic effects of MP. Treatment of polystyrene MP (5 μm) enhance total histopathological Suzuki score such as sever vacuolization and hepatocellular edema in liver tissue of C57BL/6 mice, while different ones (5 and 20 μm) induce the increase of inflammation and lipid droplets in liver tissue of ICR mice [12,19]. In the kidney, the increase of creatinine level and tubular injury were observed in C57BL/6 mice after treatment of polystyrene MP (2 μm) [16]. In reproductive organs, polystyrene MP (5 μm) induce a significant reproductive toxicity including the decrease of sperm survival rate and testis weight, disorderly arrangement of spermatid cells in the testicular seminiferous tubules, and sperm denegation in the testis tubules of ICR mice [56]. Also, a similar reproductive toxicity such as apoptosis of granulosa cells and ovary fibrosis were observed in ovary of Wistar rats treated with polystyrene MP (0.5 μm) [57]. However, the present study has been focused the characterization of constipation phenotypes in ICR mice after treatment of polystyrene MP with 0.5 μm size without analyzing the toxicity and microbiota regulatory effects of MP. Therefore, further research will be needed on the role and action mechanism of MP in toxic effects and microbiota composition.

## 4. Materials and Methods

### 4.1. Characterization of MP

Aqueous suspension of concentration 25 mg/mL MP was purchased from Sigma-Aldrich Co. (St. Louis, MO, USA), having mean particle size 0.5 μm, and density 1.04–1.06 g/cm^3^. The morphology was analyzed by SEM/EDX spectroscopy (JEOL Ltd., Tokyo, Japan), and actual size was measured by the Zetasizer Nano ZS90 (Malvern Instruments Inc., Malvern, UK). All suspensions were thoroughly dispersed by sonication, and diluted with water before use. 

### 4.2. Experimental Design of Animal Study

The animal protocol to characterize the constipation phenotype was reviewed and approved by the Pusan National University-Institutional Animal Care and Use Committee (PNU-IACUC) based on the ethical procedures for scientific care (Approval Number PNU-2020-2654, 24 June 2020). All ICR mice were maintained at the Pusan National University-Laboratory Animal Resources Center, accredited by the Korea Food and Drug Administration (KFDA) (Accredited Unit Number-000231) and The Association for Assessment and Accreditation of Laboratory Animal Care (AAALAC) International (Accredited Unit Number; 001525). All mice were provided ad libitum access to a standard irradiated chow diet (Samtako BioKorea Inc., Osan, Korea) and water. Throughout the experiment, mice were maintained in a specific pathogen-free (SPF) state under a strict light cycle (on at 08:00 h; off at 20:00 h) at 23 ± 2 °C and 50 ± 10% relative humidity.

Briefly, 7-week-old ICR mice (*n* = 24) were assigned to either a 1× PBS treated group (Vehicle, *n* = 6) or MP treated group (MP, *n* = 18). The optimal dosage for MP administration used in the animal model was decided based on results from previous studies for tissue accumulation [12], and effects on gut microbiota dysbiosis and hepatic lipid metabolism disorder [19]. The MP treated group was further divided into a low concentration MP treated group (LoMP, *n* = 6), medium concentration MP treated group (MiMP, *n* = 6), and high concentration MPs treated group (HiMP, *n* = 6). The three MP treated groups were orally administrated varying concentrations of dispersed MP solution (10 μg/L, 50 μg/L and 100 μg/L) once daily (0.5 mL/day), while the Vehicle treated group was administered the same volume of 1× PBS solution. The physiological condition of all mice in each group was regularly monitored at 10 a.m. every day during the experimental periods; there were no occurrences of severely ill or dead animals. At 2 weeks after MP administration, total stools, urine, food, and water were collected from the metabolic cage of each group for further analyses. All mice were subsequently euthanized using CO_2_ gas, after which the mid colon and serum samples were acquired and stored at −70 °C in Eppendorf tubes until assay.

### 4.3. Measurement of Food Intake and Water Consumption

Throughout the experimental duration, the food weight and water volume were measured daily in the Vehicle, LoMP, MiMP, and HiMP treated groups at 10:00 a.m., using an electrical balance and a measuring cylinder, respectively. All measurements were performed twice to ensure accuracy, and average food intake and water consumption were calculated using the measured data.

### 4.4. Analyses of Stool Parameters

Mice of subset groups were bred in individual metabolic cages (Daejong Ltd., Seoul, Korea) for 12 h, to avoid any contamination of stools and urine. Stools excreted from each mouse were collected at 10:00 a.m. Each stool weight was measured three times using an electric balance (Mettler Toledo, Columbus, OH, USA), whereas the total number of stools was counted twice per animal. The stool moisture content was determined as follows:Stool moisture content = (A − B)/A × 100(1)
where, A is the weight of fresh stools collected after administration of microplastics, and B is the weight of stools after drying at 60 °C for 24 h. The morphological image of total stools from each mouse was acquired using a digital camera, and abnormal shaped stools were sequentially counted in duplicate. Furthermore, urine volume was collected at 9 a.m. next day, and measured two times per sample, using a measuring cylinder.

### 4.5. Measurement of Gastrointestinal (GI) Transit Ratio and Intestinal Length

The GI transit ratio was measured by applying the method described previously [58]. Briefly, all experimental mice were fed 1 mL of charcoal meal (3% suspension of activated charcoal in 0.5% aqueous methylcellulose) (Sigma-Aldrich Co.); 30 min after administration, the mice were euthanized using CO_2_, and the intestinal tract was collected from the abdominal cavity. Intestinal charcoal transit ratio was calculated as follows:Charcoal transit ratio (%) = [(total small intestine length − transit distance of charcoal meal)/total small intestine length] × 100(2)

The total intestinal length was also measured from stomach to anus, in duplicate.

### 4.6. Histopathological Analysis

Mid colons collected from the Vehicle, LoMP, MiMP, and HiMP groups were fixed in 10% formalin for 48 h. Tissue samples were subsequently embedded in paraffin wax, after which they were cut into 4 μm thick sections and stained with H&E (Sigma-Aldrich Co.). The sections were subsequently analyzed by light microscopy for mucosal thickness, flat luminal surface thickness, and number of goblet cells in mid colons, applying the Leica Application Suite (Leica Microsystems Ltd., Heerbrugg, Switzerland).

Mucin staining analysis was achieved by fixing the mid colons collected from mice of all subset groups in 10% formalin for 48 h, then embedding the samples in paraffin wax and sectioning into 4 μm thick slices, that were subsequently deparaffinized with xylene and rehydrated. The mounted tissue sections were rinsed with distilled water and stained using an Alcian Blue Stain kit (IHC WORLD, Woodstock, MD, USA), after which the histological features in stained colon sections were observed by light microscopy.

### 4.7. TEM Analysis

Mid colon tissues collected from mice of subset groups were fixed in 2.5% glutaraldehyde solution, rinsed with 1× PBS solution, dehydrated with ascending concentrations of EtOH solution, post-fixed in 1% osmium tetroxide (O_s_O_4_) for 1–2 h at room temperature, and embedded in Epon-812 media (Polysciences Inc., Hirschberg an der Bergstrasse, Germany). Subsequently, ultra-thin sections of the mid colon tissue (70 nm thick) were placed on holey formvar-carbon coated grids, after which the grids were subjected to negative staining using uranyl acetate and lead citrate. Ultrastructure and distribution of Lieberkuhn crypts in mid colon were examined using the TEM (Hitachi Co., Ltd., Tokyo, Japan).

### 4.8. Western Blotting Analysis

The Pro-Prep Protein Extraction Solution (Intron Biotechnology Inc., Seongnam, Korea) was used to prepare total proteins from mid colons and IEC18 cells of Vehicle and LoMP, MiMP, HiMP treated groups, according to the manufacturer’s protocol. Protein homogenates were subsequently centrifuged at 13,000 rpm at 4 °C for 5 min, after which total protein concentrations were determined using a SMARTTM Bicinchoninic Acid Protein assay kit (Thermo Fisher Scientific Inc., Wilmington, MA, USA). Total proteins (30 μg) were subjected to 4–20% sodium dodecyl sulfate-polyacrylamide gel electrophoresis (SDS-PAGE) for 3 h, and the resolved proteins were transferred to nitrocellulose membranes for 2 h at 40 V. The membranes were then probed with the following primary antibodies, overnight at 4 °C: anti-Gα (Abcam, Cambridge, UK), anti-mAChR M2 (Alomone Labs, Jerusalem, Israel), anti-mAChR M3 (Alomone Labs), anti-PKC (Cell Signaling Technology Inc., Cambridge, MA, USA.), anti-p-PKC (Cell Signaling Technology Inc.), anti-PI3K (Cell Signaling Technology Inc.), anti-p-PI3K (Cell Signaling Technology Inc.), anti-ERK 1/2 (Cell Signaling Technology Inc.), anti-p-ERK (E-4)(Santa Cruz Biotechnology Inc., Dallas, TX, USA), anti-p38 (Cell Signaling Technology Inc.), anti-p-p38 (Cell Signaling Technology Inc.), anti- pNF-κB (Boster Bio, CA, USA), anti-IκB-α (Cell Signaling Technology Inc.), anti- p-IκB- α (Cell Signaling Technology Inc.), or anti-β-actin (Sigma-Aldrich Co.). Membranes were subsequently washed with washing buffer (137 mM NaCl, 2.7 mM KCl, 10 mM Na_2_HPO_4_, 2 mM KH_2_PO_4_, and 0.05% Tween 20), followed by incubation with 1:1000 diluted horseradish peroxidase-conjugated goat anti-rabbit IgG (Zymed Laboratories, South San Francisco, CA, USA) for 2 h at room temperature, after which the blots were developed using a Chemiluminescence Reagent Plus kit (Pfizer Inc., Gladstone, NJ, USA). Signal images of each protein were subsequently acquired using a digital camera (1.92 MP resolution) of the FluorChem^®^ FC2 Imaging system (Alpha Innotech Corporation, San Leandro, CA, USA). Protein densities were semi-quantified using the AlphaView Program, version 3.2.2 (Cell Biosciences Inc., Santa Clara, CA, USA).

### 4.9. Quantitative Realtime—Polymerase Chain Reaction (RT-qPCR) Analysis

Frozen mid colon tissue and IEC18 cells were homogenized in RNA Bee solution (Tet-Test, Friendswood, TX, USA). Total RNA molecules were isolated by centrifugation at 15,000 rpm for 15 min, after which RNA concentration was measured by the Nano Drop Spectrophotometer (Allsheng, Hangzhou, China). About 5 µg of total RNA was annealed with 500 ng of oligo-dT primer (Thermo Fisher Scientific Inc.) at 70 °C for 10 min. Complementary DNA (cDNA) was synthesized using the Invitrogen Superscript II reverse transcriptase (Thermo Fisher Scientific Inc.). qPCR was performed with the cDNA template obtained (2 µL) and 2× Power SYBR Green (6 µL; Toyobo Life Science, Osaka, Japan) containing specific primers as follows: AQP3 sense primer 5′-GGTGG TCCTG GTCAT TGGAA-3′ and antisense primer 5′-AGTCA CGGGC AGGGT TGA-3′; AQP8 sense primer 5′-TCGCT GGCAG TCACA GTGA-3′ and antisense primer 5′-TCCAA ATAGC TGGGA GATCC A-3′; MUC1 sense primer 5′-CGCCA GCCTT GAGTT TGTTT-3′ and antisense primer 5′-GAAGA AAGGA GCCCG AATGC-3′; MUC2 sense primer 5′-GCACA TTCCT TCGCA TCTTA AA-3′ and antisense primer 5′-AAAGC AAAGA ATGGA ACAGA AACTC-3′; Klf4 sense primer 5′-GGTGC AGCTT GCAGC AGTAA-3′ and antisense primer 5′-AAGTC TAGGT CCAGG AGGTC GTT-3′; CIC-2 sense primer 5′- CAGCA CATGC AAAAG CTAAG AAAA -3′ and antisense primer 5′- GCGGA TAGAT GTCTC GGAGC TA -3′; CFTR sense primer 5′- TCTGC CGCGC AGCAA -3′ and antisense primer 5′- GGTGT GAACG TCATC AGATC CA-3′; β-actin sense and antisense primers 5′-ACGGC CAGGT CATCA CTATT G-3′ and 5′-CAAGA AGGAA GGCTG GAAAA GA-3′, respectively. qPCR was performed for 40 cycles using the following sequence: denaturation at 95 °C for 15 s, followed by annealing and extension at 70 °C for 60 s. Fluorescence intensity was measured at the end of the extension phase of each cycle. Threshold value for fluorescence intensities of all samples was set manually. The reaction cycle at which the PCR products exceeded this fluorescence intensity threshold during the exponential phase of PCR amplification was considered as the threshold cycle (Ct). Expression of the target gene was quantified relative to the housekeeping gene β-actin, based on a comparison of the Cts at constant fluorescence intensity, as per the Livak and Schmittgen’s method [59].

### 4.10. Measurement of GI Hormone Concentrations

The concentrations of CCK and gastrin were quantified using ELISA kits (Cusabio Biotech Co., Ltd., Wuhan, China), according to the manufacturer’s instructions. Briefly, mid colon tissues (50 mg) were homogenized in ice-cold 1× PBS (pH 7.2–7.4) using a glass homogenizer (Sigma-Aldrich Co.). Resultant tissue lysates were then centrifuged at 1000× *g* for 5 min at 4 °C, after which the supernatant was collected for analysis. Specific antibodies for the two hormones (separately in each well) were added to the supernatant, with subsequent incubation for 1 h at 37 °C, after which HRP-Streptavidin solution was added to the mixture and further incubated for 1 h at 37 °C. This was followed by addition of the TMP One-Step Substrate Reagent and incubation for 30 min at 37 °C. The reaction was terminated by addition of the stop solution. Finally, absorbance of the reaction mixture was read at 450 nm using the VersaMax Plate Reader (Molecular Devices, Sunnyvale, CA, USA).

### 4.11. Measurement of Chloride Ion Concentration

The concentration of chloride ions in mid colons was quantified using the chloride assay kit (Abcam Co.), according to the manufacturer’s instructions. Briefly, mid colon tissue (10 mg) was homogenized in ice-cold 1× PBS (pH 7.2–7.4) using a glass homogenizer (Sigma-Aldrich Co.). Resultant tissue lysates were then centrifuged at 13,000 rpm for 5 min at 4 °C, after which the supernatant was collected for analysis. After addition of chloride reagent (separately in each well), the supernatant was incubated for 15 min at room temperature. Finally, absorbance of the reaction mixture was read at 620 nm using the VersaMax Plate Reader (Molecular Devices).

### 4.12. Cell Culture and MP Treatment

IEC18 cells, intestinal epithelioid cell line, were purchased from ATCC (Manassas, VA, USA). They were grown in Dulbecco’s modified Eagle’s medium (DMEM, Welgene, Gyeongsansi, Korea) supplemented with 10% fetal bovine serum, 2 mM glutamine, 100 U/mL penicillin, and 100 μg/mL streptomycin at 37 °C in a humidified atmosphere containing 5% CO_2_. After reaching 70–80% confluence, IEC18 cells were classified into four different groups; Vehicle, LoMP, MiMP, and HiMP treated group. They were exposed to 10 μg/mL (LoMP), 50 μg/mL (MiMP) and 100 μg/mL (HiMP) for 24 h, while Vehicle treated group was received with dH_2_O of same volume. The cell morphology was also observed under a microscope (Leica Microsystems) at 100× and 200× magnification. After then, total cells of each group were harvested for western blot and RT-qPCR analyses.

### 4.13. Statistical Analysis

Statistical significance was evaluated using the One-way Analysis of Variance (ANOVA) (SPSS for Windows, Release 10.10, Standard Version, Chicago, IL, USA), followed by Tukey’s post hoc t-test for multiple comparisons. All values are expressed as the means ± SD, and a p-value (*p* < 0.05) is considered statistically significant.

## 5. Conclusions

Taken together, results from the current study determines newly characterized constipation phenotypes in ICR mice orally administrated MP for 2 weeks, including a decrease in stool parameters, delay of gastrointestinal transit, alteration of the histopathological structure of the mid colon, and suppression of mucin. In particular, these data provide novel evidence that MP-induced constipation is tightly correlated with dysregulation of the mAChR signaling pathway, as well as chloride ion and water membrane transportation (Figure 12). We therefore conclude that our findings establish that MP can be considered as one of novel causes for chronic constipation. In addition, this study has some limitations that did not analyze action mechanism of MP on the decrease of GI transit ratio as well as not provide any significant evidence for human studies to support the finding of our study.

## Figures and Tables

**Figure 1 ijms-22-05845-f001:**
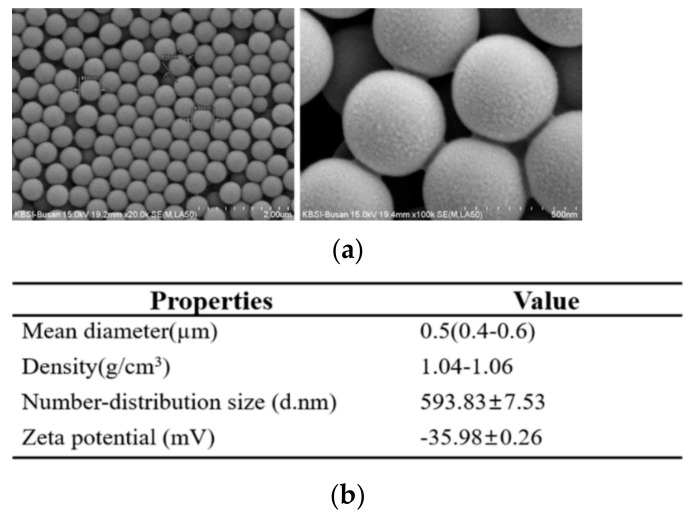
Morphology and physicochemical properties of MP. (**a**) Morphological properties of MP were observed with scanning electron microscopy (SEM), as described in Materials and Methods. (**b**) The physicochemical properties of MP were analyzed by applying methods described in previ-ous studies. Data are reported as the mean ± SD.

**Figure 2 ijms-22-05845-f002:**
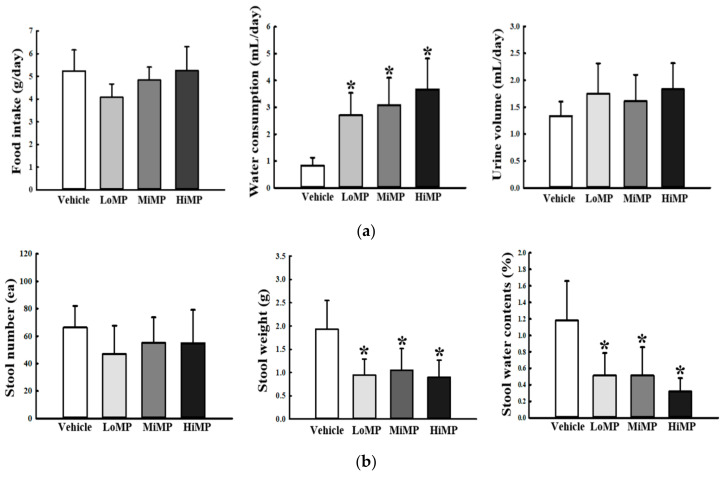
Feeding behavior, stool parameters. (**a**) Food intake and water consumption were calculated by measuring the amount of feed (water) supplied and the amount of feed (water) remaining. (**b**) Total number and weight of stools were measured as described in Materials and Methods. Stool water content was calculated using the weight of fresh stools and dried weight. (**c**) Stool morphological characteristics. Digital camera images of stools were taken immediately after collection from the metabolic cage. Four to six mice per group were used for food, water, urine, and stool collection, and each parameter was assayed in duplicate. The data are reported as the mean ± SD. *, *p* < 0.05 compared to the Vehicle treated group. Abbreviation: LoMP, Low concentration of microplastics; MiMP, Medium concentration of microplastics; HiMP, High concentration of microplastics.

**Figure 3 ijms-22-05845-f003:**
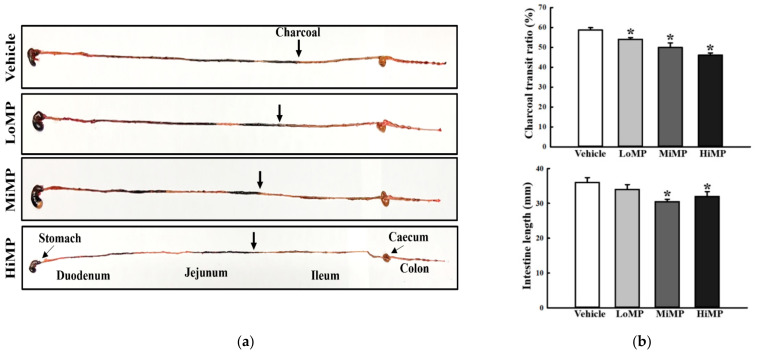
GI transit ratio and intestinal length. (**a**) Actual image showing the charcoal meal transit and intestine. The total intestinal tract was excised from mice of each subset group treated with charcoal meal powder. Morphology was observed using a digital camera. The arrows indicate position of the charcoal meal. (**b**) Transit ratio of the charcoal meal and the length of intestine. The total distance travelled by the charcoal meal from the pylorus was measured. The charcoal meal transit ratio was then calculated using total length of the intestine and distance of the charcoal meal. Four to six mice per group were used in the GI transit ratio test, and the charcoal meal transit distance and intestine length were measured in duplicate. The data are reported as the mean ± SD. *, *p* < 0.05 compared to the Vehicle treated group. Abbreviation: LoMP, Low concentration of microplastics; MiMP, Medium concentration of microplastics; HiMP, High concentration of microplastics.

**Figure 4 ijms-22-05845-f004:**
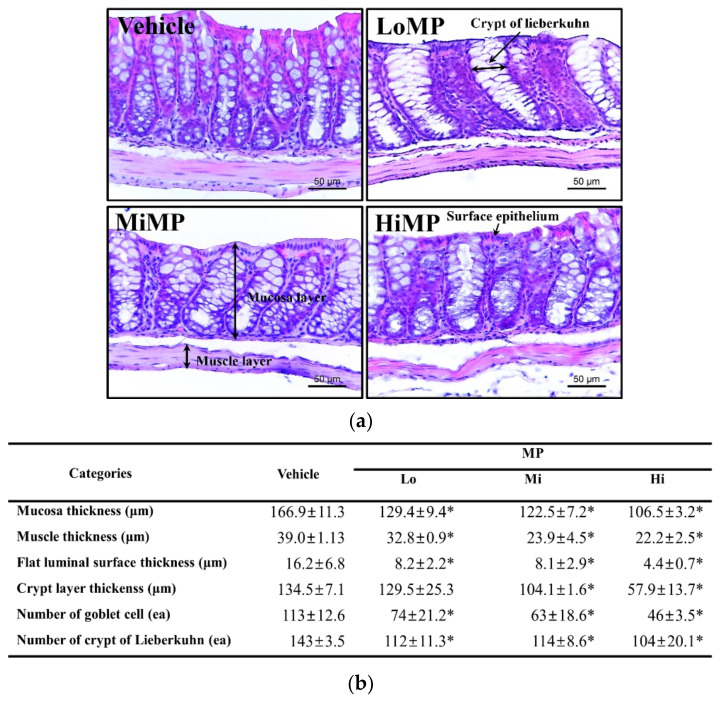
Histopathological structures of the mid colon. (**a**) H&E-stained sections of mid colon from the Vehicle, LoMP, MiMP, and HiMP treated groups were observed at 400× magnification using a light microscope. (**b**) Histopathological parameters were determined using the Leica Application Suite. Four to six mice per group were used in the preparation of H&E-stained slides, and the histopathological parameters were measured in duplicate for each slide. Data are reported as the mean ± SD. *, *p* < 0.05 compared to the Vehicle treated group. Abbreviation: LoMP, Low concentration of microplastics; MiMP, Medium concentration of microplastics; HiMP, High concentration of microplastics.

**Figure 5 ijms-22-05845-f005:**
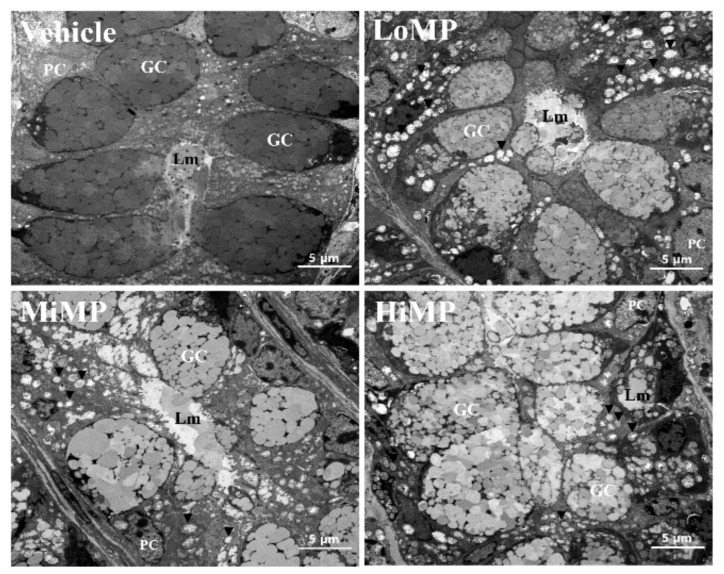
TEM images of the mid colon. The crypt ultrastructure of the mid colon in the Vehicle, LoMP, MiMP, and HiMP treated groups was observed by TEM at 4000× magnification. Two to three mice per group were used to prepare the TEM slides, and three parameters were assayed in duplicate in each test. The vesicle around the lumen is indicated by the arrowhead. Abbreviation: LoMP, Low concentration of microplastics; MiMP, Medium concentration of microplastics; HiMP, High concentration of microplastics; GC, goblet cells; PC, Paneth cell; Lm, lumen.

**Figure 6 ijms-22-05845-f006:**
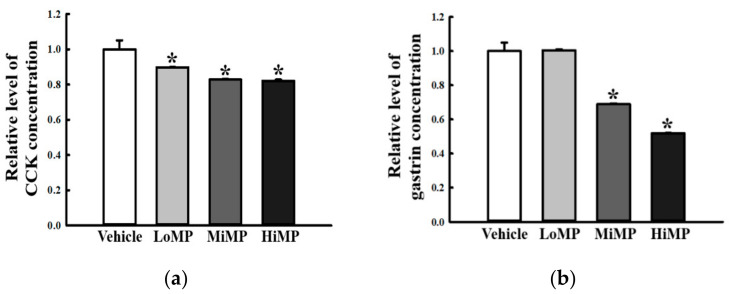
Concentrations of GI hormones. The concentrations of (**a**) CCK and (**b**) gastrin were measured in the mid colon homogenate by Enzyme-Linked Immunosorbent Assay. The minimum detectable concentration of each kit is 0.1–1000 pg/mL of CCK, and 0.312–20 pg/mL of gastrin. Five to six mice per group were used in the preparation of tissue homogenate, and hormone levels were assayed in duplicate for each sample. Data are reported as the mean ± SD. *, *p* < 0.05 compared to the Vehicle treated group. Abbreviations: LoMP, Low concentration of microplastics; MiMP, Medium concentration of microplastics; HiMP, High concentration of microplastics; GI, Gastrointestinal; CCK, Cholecystokinin.

**Figure 7 ijms-22-05845-f007:**
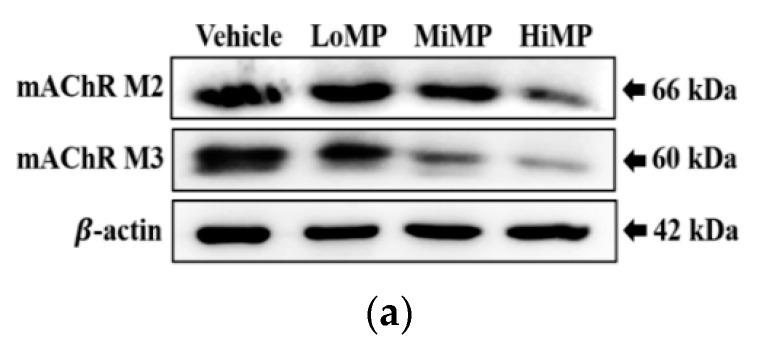
Expressions of mAChRs and key mediators within their downstream signaling pathway. (**a**,**b**) Expression levels of mAChR M2 and M3 were measured in the mid colon by Western blot analysis using specific primary antibodies and HRP-labeled anti-rabbit IgG antibody. (**c**,**d**) Expression levels of Gα, PKC, p-PKC, PI3K, and p-PI3K in the mAChR M2 and M3 signaling pathway were measured by Western blot analysis using specific primary antibodies and HRP-labeled anti-rabbit IgG antibody. After the intensity of each band was determined using an imaging densitometer, relative levels of the four proteins were calculated based on the intensity of β-actin. Four to six mice per group were used in the preparation of the total tissue homogenate, and Western blot analyses were assayed in duplicate for each sample. Data are reported as the mean ± SD. *, *p* < 0.05 compared to the Vehicle treated group. Abbreviations: LoMP, Low concentration of microplastics; MiMP, Medium concentration of microplastics; HiMP, High concentration of microplastics; mAChR, muscarinic acetylcholine receptors; PKC, Protein kinase C; PI3K, Phosphoinositide 3-kinases; HRP, Horseradish peroxidase; IgG, Immunoglobulin G.

**Figure 8 ijms-22-05845-f008:**
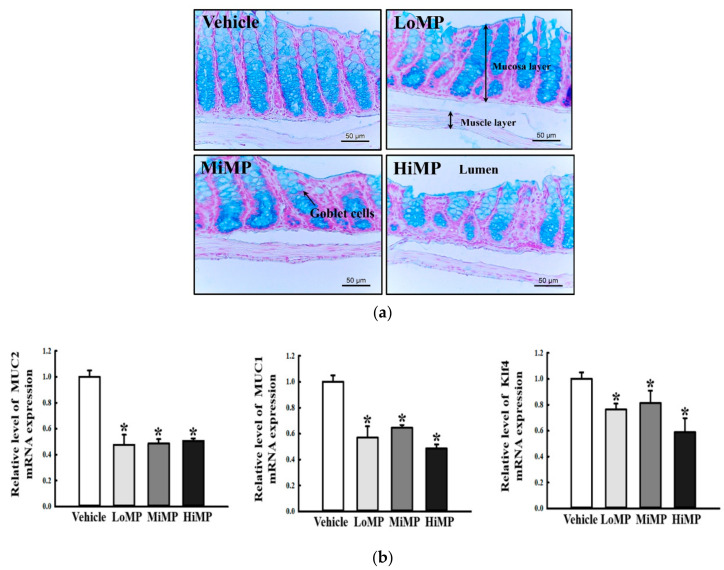
Secretion and production of mucin. (**a**) Mucin secreted from the crypt layer cells was stained with Alcian blue at pH 2.5, and images were observed at 100× magnification. Four to six mice per group were used in the preparation of tissue slides, and Alcian blue staining analysis was performed in duplicate for each slide. (**b**) The levels of MUC2, MUC1, and Klf4 transcripts in the total mRNA of mid colons were measured by RT-qPCR using specific primers. The mRNA levels of the three genes were calculated, based on the intensity of actin as an endogenous control. Four to six mice per group were used in the preparation of total RNA, and RT-qPCR analyses were assayed in duplicate for each sample. The data are reported as the mean ± SD. *, *p* < 0.05 compared to the Vehicle treated group. Abbreviations: LoMP, Low concentration of microplastics; MiMP, Medium concentration of microplastics; HiMP, High concentration of microplastics; RT-qPCR, Quantitative real time-PCR; MUC2, Mucin 2; MUC1, Mucin 1; Klf4, Kruppel Like Factor 4.

**Figure 9 ijms-22-05845-f009:**
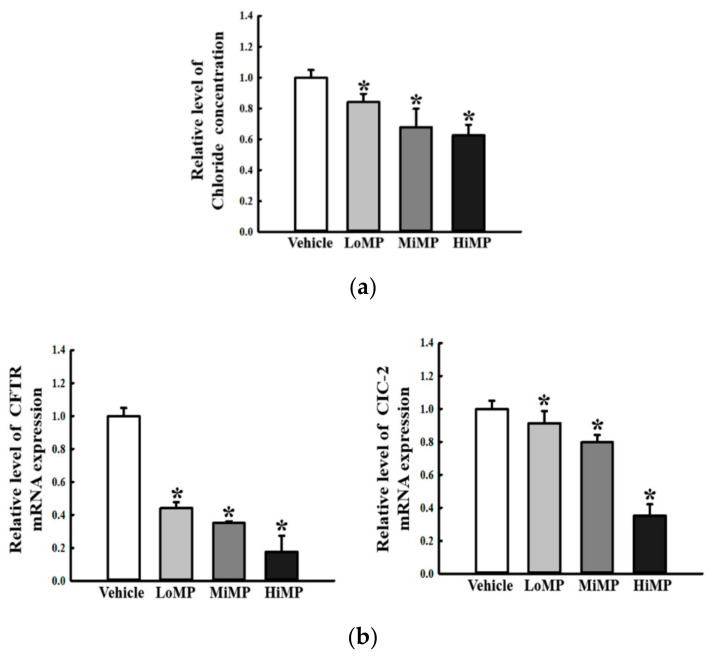
Concentration of chloride ion and its channel expression. (**a**) The concentration of chloride ion was determined in the mid colon using the chloride assay kit. (**b**) The levels of CFTR and CIC2 transcripts in the total mRNA of mid colons were measured by RT-qPCR using specific primers. The mRNA levels of the three genes were calculated, based on the intensity of actin as an endogenous control. Four to six mice per group were used the preparation of total RNA; RT-qPCR analyses were assayed in duplicate for each sample. Data are reported as the mean ± SD. *, *p* < 0.05 compared to the Vehicle treated group. Abbreviations: LoMP, Low concentration of microplastics; MiMP, Medium concentration of microplastics; HiMP, High concentration of microplastics; RT-qPCR, Quantitative real time-PCR; CFTR, CIC-2, Chloride channel 2; Cystic fibrosis transmembrane conductance regulator.

**Figure 10 ijms-22-05845-f010:**
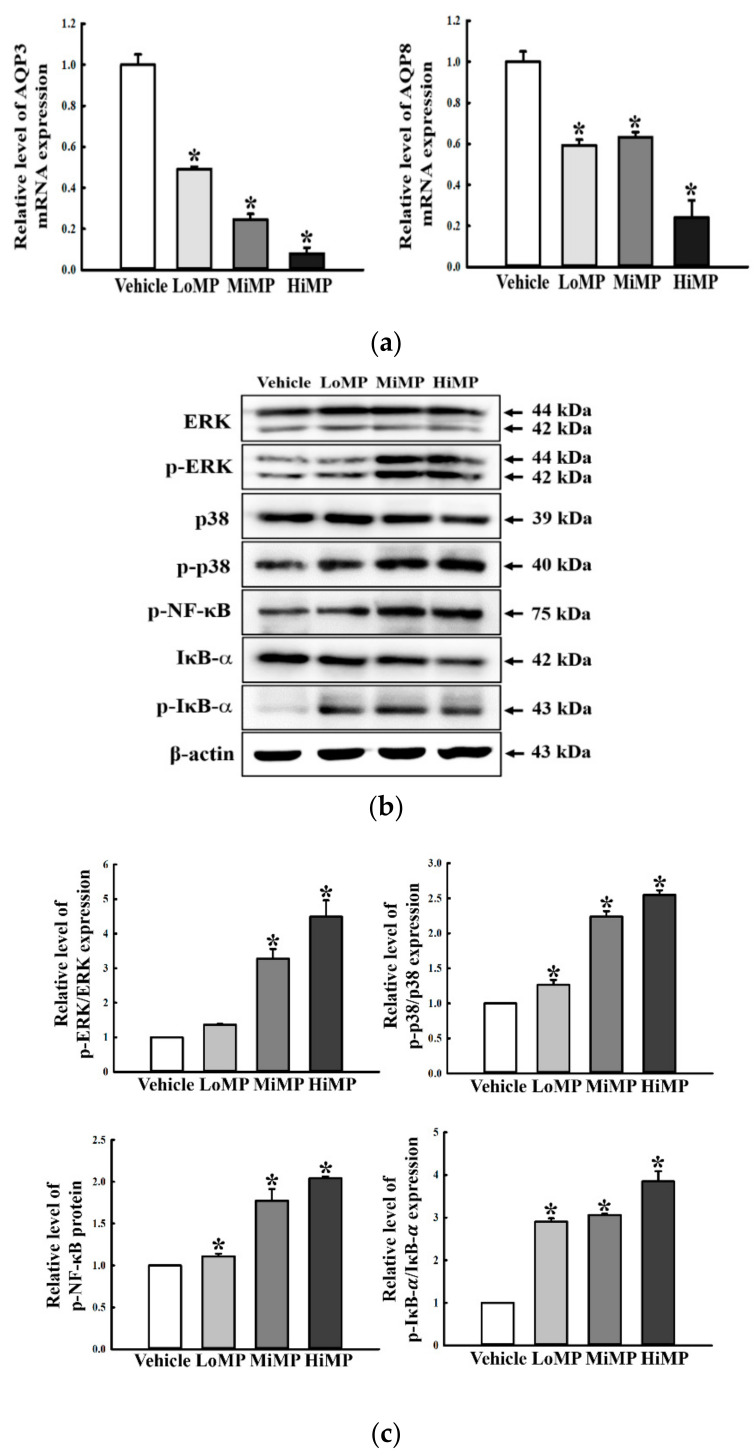
The Expressions of AQP and key mediators of its downstream signaling pathway. (**a**) The levels of AQP3 and 8 transcripts in the total mRNA of mid colons were measured by RT-qPCR using specific primers. The mRNA levels of the three genes were calculated, based on the intensity of actin as an endogenous control. Four to six mice per group were used the preparation of total RNA; RT-qPCR analyses were assayed in duplicate for each sample. (**b**,**c**) Expression levels of ERK, p-ERK, p38, p-p38, p-NF-κB, IκB-α, and p-IκB-α in the MAPK/NF-κB signaling pathway were measured by Western blot analysis using specific primary antibodies and HRP-labeled anti-rabbit IgG antibody. After the intensity of each band was determined using an imaging densitometer, relative levels of the four proteins were calculated based on the intensity of actin. Four to six mice per group were used in the preparation of the total tissue homogenate, and Western blot analyses were assayed in duplicate for each sample. Data are reported as the mean ± SD. *, *p* < 0.05 compared to the Vehicle treated group. Abbreviations: LoMP, Low concentration of microplastics; MiMP, Medium concentration of microplastics; HiMP, High concentration of microplastics; ERK, Extracellular-signal-regulated kinase; NF-κB, Nuclear factor κB; IκB-α, inhibitor of κB-α.

**Figure 11 ijms-22-05845-f011:**
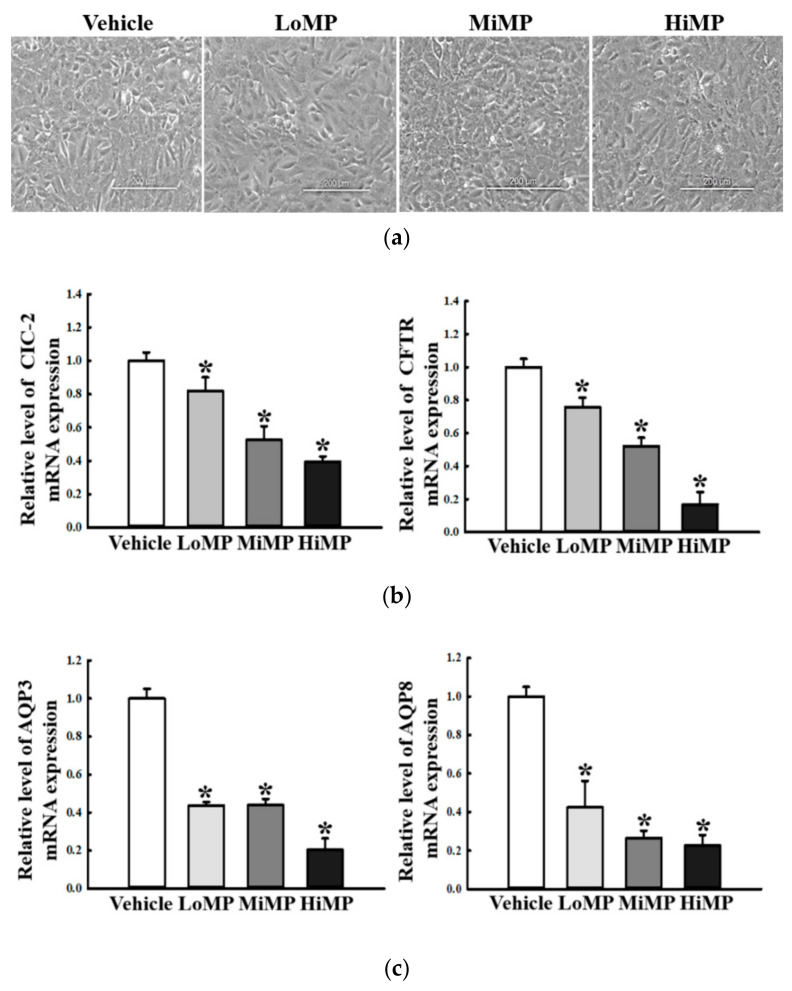
Expressions of chloride channel, AQP and key mediators of its downstream signaling pathway in IEC18 cells. (**a**) IEC18 cells were treated with 10 μg/mL (LoMP), 50 μg/mL (MiMP), and 100 μg/mL (HiMP) for 24 h. Cell morphologies were observed under a microscope at 100× magnification. (**b**) The levels of CFTR and CIC2 transcripts in the total mRNA of IEC18 cells were measured by RT-qPCR using specific primers. (**c**) The levels of AQP3 and 8 transcripts in the total mRNA of IEC18 cells were measured by RT-qPCR using specific primers. The mRNA levels of the four genes were calculated, based on the intensity of actin as an endogenous control. Four to six mice per group were used the preparation of total RNA; RT-qPCR analyses were assayed in duplicate for each sample. (**d**) Expression levels of ERK, p-ERK, p38, p-p38, p-NF-κB, IκB-α, and p-IκB-α in the MAPK/NF-κB signaling pathway were measured by Western blot analysis using specific primary antibodies and HRP-labeled anti-rabbit IgG antibody. (**e**) After the intensity of each band was determined using an imaging densitometer, relative levels of the four proteins were calculated based on the intensity of actin. Four to six mice per group were used in the preparation of the total tissue homogenate, and Western blot analyses were assayed in duplicate for each sample. Data are reported as the mean ± SD. *, *p* < 0.05 compared to the Vehicle treated group. Abbreviations: LoMP, Low concentration of microplastics; MiMP, Medium concentration of microplastics; HiMP, High concentration of microplastics; ERK, Extracellular-signal-regulated kinase; NF-κB, Nuclear factor κB.

**Figure 12 ijms-22-05845-f012:**
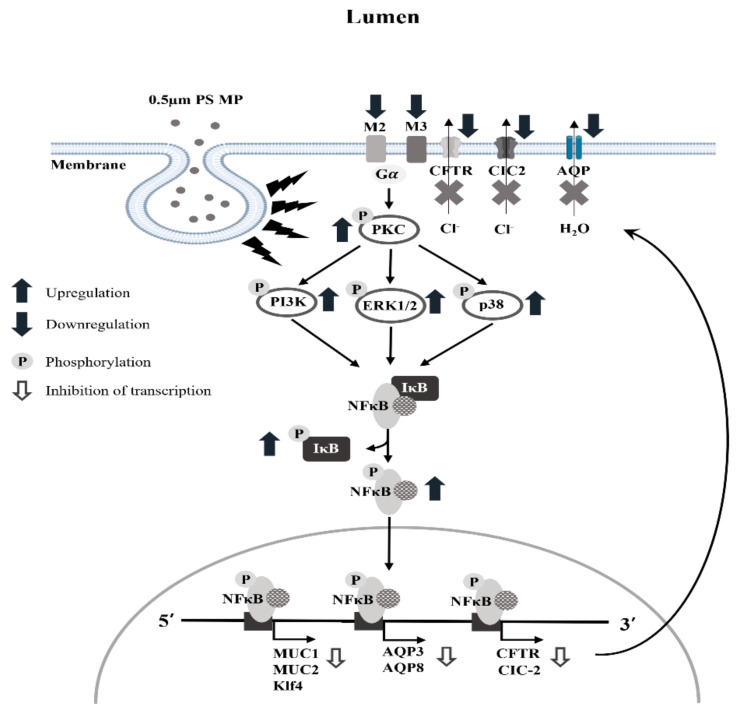
Suggested mechanism of MP-induced constipation in ICR mice. In this scheme, the internalization of PS-MP is thought to be affected by the mAChRs downstream signaling pathway through the regulation of PKC, MAPK and NF-κB. Finally, the activated NF-κB translocate into the nucleus and inhibits the expression of mucin, AQP and chloride ion channel genes.

## Data Availability

All the data that support the findings of this study are available on request from the corresponding author.

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
