# Peer review of "Novel Characterization of Constipation Phenotypes in ICR Mice Orally Administrated with Polystyrene Microplastics"

_ijms, 2021, doi:10.3390/ijms22115845_

Round 1

Reviewer 1 Report

The study is interesting and I have the following suggestions:

1, How is the dosage determined? Why did the authors choose 10 ug/L, 50 ug/L and 100 ug/L once daily? Are there any clinical studies to support this? 

2, Are there any human studies to support the findings of the present study? 

3, How MP induced a decrease of GI transit ratio? 

4, What is the precise target of the toxic effects of MP? Could microbiota play a role here? This should be discussed in the manuscript. 

Reviewer 2 Report

Choi et al studied the effect of orally administrated with polystyrene microplastics on induction of constipation in ICR mice. The authors showed that  there are alterations in water consumption, stool weight, stool water 
contents, and stool morphology were detected in the three groups of MP treated ICR mice (low, middle, and high), as compared to Vehicle treated group.  In addition, the GIT motility , intestinal length, the histopathological structure and cytological structure of the transverse colon were changed in MP  treated mice compared to the control group. Finally the authors investigated the mechanistic pathway by which MP can induce constipation by studying MAPK/NF-B signaling pathway.

Overall the manuscript is written well, However, there are several questions regarding the manuscript

General comments:

1- How did the authors determine the doses of MP ((10 ug/L, 50 ug/L and 100 ug/L) as low, middle, high? or why the authors choose these doses for the study?

2- What is the transverse colon in the mice that authors analyzed in the manuscript from Fig 4 till the end of experiment?

3- The quality of the figures is not good in some figures that I can not judge the results or authors' interpretation for examples : figure 2c, Figure 3a, Figure 4a, Figure 8a. The authors should provide high quality figures.

4- The results of Western blot is weird.  In Figure 10, the authors measured the expression of the following proteins (ERK,p38,NF-kB, and IkB-a) and the phosph form

a) Did the author perform WB for NF-kB?

b) Why the authors normalized the level of phosph protein to actin, not to the total protein?

c) According to the figure p-p38 was increased in MP treated mice compared to control mice, while total p38 was in the inverse direction. How can the authors explain these results?

d) Similarly, p-pIkB-a was increased in MP treated mice compared to control mice, while total pIkB-a  was in the inverse direction. How can the authors explain these results?

e) The authors showed only band intensity for phosph protein not to total protein. Why???

5) The same comment for Figure 11 as for Figure 10

Round 2

Reviewer 1 Report

The authors have addressed my major concerns. I suggest to accept this manuscript. 

Reviewer 2 Report

I checked carefully the replies of the authors to my commnets. The authors ansewered adequately to some questions. However, they do not give statistfactory response to other questions

a) The qulaity of the figures is till poor. Please try to replcae with high quality figures especially : Fig 3a, Fig 4a, Fig 8a, Fig 11 especially pikB-a panel.

b) What is the transvesre colon region in mouse? According to the colon anatamoy:

Human colon:  ascending, trasnverse, descending and cecum.

Mouse colon: proximal, mid, and distal. 

So where is the trasverse colon region that the authors talked about in the manuscript present in the mouse colon?????? This was my question in the previous review.

Round 3

Reviewer 2 Report

I asked about the quality of the figures. I understand that the authors replaced old figures with new ones. However, the new figures are still not of good quality.

Anyhow, I do not have further concerns about the manuscript and I guess the production can improve the quality of existing figures.